# The Effect of Restoration Thickness on the Fracture Resistance of 5 mol% Yttria-Containing Zirconia Crowns

**DOI:** 10.3390/ma17020365

**Published:** 2024-01-11

**Authors:** Po-Hsu Chen, Esra Elamin, Akram Sayed Ahmed, Daniel A. Givan, Chin-Chuan Fu, Nathaniel C. Lawson

**Affiliations:** 1Division of Prosthodontics, University of Alabama at Birmingham School of Dentistry, Birmingham, AL 35209, USA; pohsu@uab.edu (P.-H.C.); esra@uab.edu (E.E.); dgivan@uab.edu (D.A.G.); ccfu@uab.edu (C.-C.F.); 2Faculty of Dentistry, Department of Dental Biomaterials, Tanta University, Tanta 31527, Egypt; akram_gad@dent.tanta.edu.eg

**Keywords:** zirconia, cementation, crown fracture

## Abstract

Background: To determine what thickness of 5 mol% yttria zirconia (5Y-Z) translucent crowns cemented with different cements and surface treatments would have equivalent fracture resistance as 3 mol% yttria (3Y-Z) crowns. Methods: The study included 0.8 mm, 1.0 mm, and 1.2 mm thickness 5Y-Z (Katana UTML) crowns and 0.5 and 1.0 mm thickness 3Y-Z (Katana HT) crowns as controls. The 5Y-Z crowns were divided among three treatment subgroups (*n* = 10/subgroup): (1) cemented using RMGIC (Rely X Luting Cement), (2) alumina particle-abraded then luted with the same cement, (3) alumina particle-abraded and cemented using a resin cement (Panavia SA Cement Universal). The 3Y-Z controls were alumina particle-abraded then cemented with RMGIC. The specimens were then loaded in compression at 30° until failure. Results: All 5Y-Z crowns (regardless of thickness or surface treatment) had a similar to or higher fracture force than the 0.5 mm 3Y-Z crowns. Only the 1.2 mm 5Y-Z crowns with resin cement showed significantly similar fracture force to the 1 mm 3Y-Z crowns. Conclusion: In order to achieve a similar fracture resistance to 0.5 mm 3Y-Z crowns cemented with RMGIC, 5Y-Z crowns may be as thin as 0.8 mm. To achieve a similar fracture resistance to 1.0 mm 3Y-Z crowns cemented with RMGIC, 5Y-Z crowns must be 1.2 mm and bonded with resin cement.

## 1. Introduction

The use of all-ceramic dental restorations has increased in the past decade due to their cost, esthetic appearance, advances in adhesive dentistry, and improvements in dental ceramic materials [1]. Compared with other high-strength ceramic materials, 3 mol% yttria-stabilized zirconia (3Y-Z) has a higher flexural strength and fracture toughness, allows a more conservative dental preparation, minimizes wear on its antagonist, and lacks the unwanted complication of chipping when used as a monolithic restoration [2,3]. Due to the inferior translucency of 3Y-Z relative to glass ceramics, more translucent zirconia has been developed [4]. Crystal structure modifications induced by an increased yttrium oxide content from 3 to 5 mol% led to the development of translucent 5 mol% yttria-stabilized zirconia polycrystal (5Y-Z). This zirconia contains a cubic crystal content of approximately 50% of the structure, which allows improved light transmission [5,6,7]. In contrast, the higher proportion of cubic crystals in 5Y-Z also contributes to its weaker flexural strength and fracture toughness due to the lack of transformation toughening that occurs in 3Y-Z [8,9].

The present ubiquity of dental zirconia necessitates clear preparation guidelines for its use. The manufacturer’s recommended minimum thickness for 3Y-Z is 0.5 mm and for 5Y-Z is 1.0 mm [10]. These proposed minimum material thicknesses are presumably based on laboratory strength testing of these materials relative to clinically proven materials. Unfortunately, the source of justification for the manufacturer’s minimum thickness values is not provided. A previous study examined the strength of 3Y-Z at different thicknesses and recommended 1.0 mm thickness to achieve an equivalent fracture resistance to a 1.5 mm metal ceramic crown [11]. To date, there has not been a previous study to verify the minimum thickness of 5Y-Z relative to a more clinically validated material, such as a 3Y-Z control.

One previous study examined the strength of 3Y-Z and 5Y-Z crowns at different thicknesses. Adabo et al. tested 1.0 and 1.5 mm thick premolar crowns fabricated from 3Y-Z and 5Y-Z that were bonded to metal dies with resin cement and fatigue-loaded with step-wise load increases [12]. The characteristic strength of the 5Y-Z crowns at both 1 and 1.5 mm was less than that of the 1 mm 3Y-Z crowns.

Several other studies examined 3Y-Z and 5Y-Z discs at different thicknesses. Alraheam et al. fatigue-loaded 0.7 and 1.2 mm thick 3Y and 5Y zirconia discs for 1.2 million cycles and 110 N of load [13]. After fatigue testing, none of the 3Y zirconia specimens of either thickness fractured, whereas 80% of the 0.7 mm thick 5Y-Z specimens and 30% of the 1.2 mm thick 5Y-Z specimens fractured. Longhini et al. studied 0.5, 1, and 1.5 mm 3Y-Z and 5Y-Z discs bonded to G10 epoxy discs with resin cement and tested them using a biaxial flexural strength apparatus [14]. The strength of the 0.5 mm thick 3Y-Z and 5Y-Z discs was statistically similar; although no statistical comparison was performed, the strength after bonding was similar between the 1.0 mm thick 3Y-Z discs and the 1.5 mm thick 5Y-Z discs. Machry et al. studied 0.7 and 1.0 mm thick discs of 3Y-Z and 5Y-Z bonded to epoxy resin, resin composite, or metal dies and fatigue-loaded with step-wise load increases [15]. The 0.7 mm 3Y-Z discs produced higher fracture resistance than the 0.7 and 1.0 mm 5Y-Z discs on the epoxy resin and resin composite dies.

Aside from the thickness of a crown, its fracture resistance may also be affected by the cement used and its surface preparation. The use of resin cement has been shown to improve the fracture resistance of both 3Y-Z and 5Y-Z crowns [16,17,18]. Regardless, resin-modified glass ionomer cement (RMGIC) is the most commonly used cement for zirconia crowns [19]. Therefore, this study employed 3Y-Z controls cemented with RMGIC and tested 5Y-Z crowns with both RMGIC and resin cement to determine the ability of resin cement to compensate for the reduced material strength of 5Y-Z.

Bonding to zirconia requires air particle abrasion and the use of a primer or resin cement containing 10-methacryloyloxydecyl dihydrogen phosphate (10-MDP) monomers [20,21]. Without air particle abrasion, the bond strength to both 3Y-Z and 5Y-Z decreases significantly [22,23,24]. The flexural strength of 3Y-Z is not negatively affected by air particle abrasion; however, 5Y-Z can be weakened following air particle abrasion [22,25,26]. With the use of RMGIC on zirconia crowns, air particle abrasion has not been shown to improve the bond as it has with resin cement [27]. Therefore air particle abrasion would not be necessary for 5Y-Z when used with RMGIC. Regardless, there may be some laboratories and clinicians who routinely air particle-abrade 5Y-Z crowns, even if RMGIC will be used for cementation [28]. Additionally, instructions for the use of some RMGICs recommend air particle abrasion on the internal surface of zirconia crowns [29]. In the current study, the recommended protocol for bonding 5Y-Z (air particle abrasion and resin cement), the recommended protocol for the conventional cementation of 5Y-Z (air particle abrasion and RMGIC), and an alternative protocol for conventional cementation (no air particle abrasion and RMGIC) were evaluated.

There are different testing methodologies to study the effect of restoration material thickness and surface treatment on the strength of dental ceramics. The International Standards Organization (ISO) standard for dental ceramic materials (ISO 6872:2015 [30]) describes three methods for the calculation of strength: three-point flexural, four-point flexure, and biaxial flexure (piston-on-three-ball). Three- and four-point tests utilize a bar supported on two ends and loaded across its center at one or two points, respectively. Biaxial flexural testing employs a disc specimen supported near its periphery and loaded in the center. Ceramics fail when the weakest flaw within the material propagates a critical crack. For this reason, four-point flexural testing leads to lower strength values than three-point flexural testing because there is a greater area subjected to the maximum bending moment (between the loading points), and there is a higher probability that a critical flaw will be present in this area [31]. Biaxial flexural testing will provide higher strength than three- or four-point flexural testing as the specimens are not loaded on their edges, where they are susceptible to failure due to flaws introduced in their fabrication process [31]. The advantage of using standardized geometry for specimens when testing the strength of materials is that specimens are consistent between testing sites, and loading may be applied such that stresses are determined by known calculations.

Another method of testing strength is the use of the crown fracture test, which utilizes a crown form cemented to a tooth preparation die that is loaded on its occlusal/lingual surface to failure. The advantages of crown fracture tests include the following. First, flaws or stress risers introduced into the crowns that result from their fabrication process can be factored into their measured strength. The method of fabrication of different ceramics may lead to specimens with different surface flaws that are more representative of actual clinical conditions than a flat bar [32]. Treating the specimens similarly to how they are used clinically ensures that testing is representative of clinical situations rather than selecting specimen preparation that is convenient for laboratory testing [33]. Despite the standardization offered by ISO testing, the ISO standard is not specific in the methods required to polish flexural strength specimens, which has led to different laboratories reporting 50% lower strength values of the same dental ceramic in round robin testing [34].

Second, the presence of a supporting structure may affect the strength of one material more than another [35]. Some materials may bond better to their substructure die, which more efficiently allows stress transfer [36]. Also, the different mismatch in elastic modulus between the crown and the substructure die may allow some materials to fare better than others [15,37]. Previous studies have reported that rankings of materials with crown fracture load testing do not correlate with flexural strength testing [38].

The objective of this study was to investigate the fracture resistance of 5Y-Z crowns at 0.8 mm, 1.0 mm, and 1.2 mm thickness that were bonded with resin cement (with air particle abrasion) or cemented with RMGIC (with and without air particle abrasion) using a crown fracture test. The values were compared to 3Y-Z cemented with its most common cement (RMGIC) at the manufacturer’s recommended thickness (0.5 mm) and a thickness which has been shown to be equivalent to a metal ceramic crown (1.0 mm). The purpose of the study is to provide clinical guidance for the preparation of 5Y-Z crowns. The two null hypotheses are that there would be no difference in the fracture resistance of any of the 5Y-Z crowns tested relative to the 0.5 mm 3Y-Z control or the 1.0 mm 3Y-Z control.

## 2. Materials and Methods

An acrylic maxillary premolar on an artificial dentiform was used to form a standardized tooth preparation of minimum height 4 mm and 1 mm margin width. A coarse diamond tapered rotary cutting bur (6856.31.016 FG Coarse Round-End Taper Diamond, Brasseler, Savannah, GA, USA) was secured on the high-speed handpiece and kept parallel to the vertical axis of the tooth to create a standardized angle of convergence (6–10°). The shape of the diamond rotary formed a modified chamfer finish line. Occlusal, anatomical reduction of 1 mm was accomplished with the same bur. The prepared tooth was scanned with a lab scanner (E3 Scanner, 3Shape Inc., Copenhagen, Denmark); then, subsequent design and model construction were accomplished through computer-aided design (CAD) software (Dental System 2020, 3Shape Inc.) for both digital die and crown fabrication (Figure 1). Crowns were fabricated using a uniform coping design with a thickness of either 0.5, 0.8, 1.0, or 1.2 mm and a 20 µm die spacer.

The uniform crowns were milled from either a 5Y-Z disk (Katana UTML, Kuraray Noritake, Tokyo, Japan) for the 0.8, 1.0, and 1.2 mm groups or the 3Y-Z disk (Katana HT, Kuraray Noritake) for the 0.5 and 1.0 mm control groups (Table 1). Milling was performed using a 5-axis milling machine (DWX-52D, Roland DGA, Irvine, CA, USA) and computer-aided manufacturing software (Millbox 2020, CIM System, Padova, Italy). After the milling procedure, final sintering of zirconia was performed according to manufacturers’ instructions.

Resin dies were fabricated using a dental 3D-printer (Pro S Dental 3D Printer, SprintRay Inc., Los Angeles, CA, USA) with a micro-filled hybrid composite resin (NextDent C&B MFH, NextDent B.V., Soesterberg, The Netherlands) with elastic modulus of 2.1 GPa [39]. After printing, the dies were cleaned in 91% alcohol and post-cured with a curing machine (SprintRay Procure 2, SprintRay Inc.).

Specimens were further divided into 3 subgroups to evaluate the cements and surface treatments utilized. The first subgroup of crowns (5Y-Z crowns only) was cemented on the resin dies using self-curing resin-modified glass ionomer cement (RMGIC; Rely X Luting Cement, 3M, St Paul, MN, USA) without surface treatment. The second subgroup of crowns (3Y-Z and 5Y-Z crowns) was air particle-abraded using 50 µm particles of Al_2_O_3_ (Cobra, Renfert, Hilzingen, Germany) at 0.2 MPa for 15 s, at a distance of 10 mm on intaglio surfaces at 30-degree incidence using a laboratory sandblaster (Basic Master, Renfert). Crowns were then cleaned ultrasonically in a distilled water bath for 10 min and dried using oil-free air, then cemented using the same RMGIC. The third subgroup of crowns (5Y-Z crowns only) was air particle-abraded and cleaned using the protocol above. Crowns were then bonded using a self-adhesive resin cement which contained the monomer 10-MDP (Panavia SA Cement Universal, Kuraray Noritake). For each group, a 10 N load was applied on the occlusal surface of all crowns, and the cement margin was tack-cured for 3–5 s per side. The excess cement was removed, and the crowns were allowed to self-cure for 7 (RMGIC) or 5 (resin cement) minutes. Specimens were stored in water at 37 °C for 7 days.

Specimens were inserted into a custom fixture in a universal testing machine (Instron 5583, Instron Inc., Canton, MA, USA) that allowed the long axis of the tooth to be positioned at a 30° angle to the indenter (Figure 2). A 3.5 mm diameter stainless steel indenter was centered in the occlusal groove of the crowns such that the crowns were loaded on their buccal cusp (Figure 2). A rubber sheet was inserted between the indenter and the crowns to account for any irregularities in the surface of the indenter. A load was applied at a crosshead speed of 0.5 mm/min until fracture. Fracture was defined as a 30% reduction in the applied load. After each test, specimens were examined to ensure that either a complete fracture or crack was present in the specimen. Fracture force was recorded as the highest load prior to fracture.

Normality of data was examined using histograms and confirmed with Shapiro–Wilk test. A one-way Analysis of Variance (ANOVA) test was employed to determine if statistical differences existed between 5Y-Z crown fracture values and the 0.5 mm 3Y-Z control using analytics software (SAS 9.4, SAS Institute, Nashville, TN, USA). A separate 1-way ANOVA was performed to determine if statistical differences existed between 5Y-Z crown fracture values and the 1 mm 3Y-Z control. Subsequent Dunnett post hoc tests were performed to determine which groups of the 5Y-Z crowns were statistically less than each control. A *p* value of less than 0.05 was considered significant.

## 3. Results

A post hoc power analysis was completed using an effect size = 0.4, a = 0.05, number of groups = 11, and sample size = 110. The G-power calculation determined a power of 80% [40]. The results of the crown fracture testing are presented in Table 2 and Figure 3. The one-way ANOVA tests determined significant differences between groups with both the 0.5 mm control (F = 33.177, *p* < 0.001) and 1.0 mm control (F = 33.250, *p* < 0.001). The Dunnett post hoc test determined that none of the 5Y-Z crown groups had a significantly lower fracture strength than the 0.5 mm 3Y-Z control. A second Dunnett post hoc test determined that all of the 5Y-Z crown groups had significantly lower fracture strength than the 1.0 mm 3Y-Z control other than the 1.2 mm 5Y-Z crowns bonded with resin cement.

Observation of failed specimens reveals that crowns fractured into two major pieces. The location of the fracture on the external aspect of the crowns was between the area of indenter contact and the occlusal groove of the crown. The crowns fractured either with or with die fracture (Figure 4a,b).

## 4. Discussion

The first aim of this study was to examine what thickness and cementation condition of 5Y-Z would produce a crown with similar fracture resistance to a 0.5 mm thick 3Y-Z cemented with RMGIC. This reference chosen as 0.5 mm is the manufacturer’s recommended thickness for 3Y-Z, and most zirconia crowns are cemented with RMGIC [11,16]. All thicknesses of 5Y-Z tested (0.8, 1.0, and 1.2 mm) achieved a similar fracture force to this reference regardless of cement type or surface treatment.

The second aim of this study was to examine what thickness and cementation condition of 5Y-Z would produce a crown with similar fracture resistance to a 1.0 mm thick 3Y-Z cemented with RMGIC. This reference chosen as 1.0 mm thickness of 3Y-Z was reported to have a similar fracture resistance as metal ceramic crowns [10]. The results of the study demonstrated that all 5Y-Z crowns (0.8, 1.0, and 1.2 mm thickness) with all cementation conditions had significantly lower fracture resistance than 1.0 mm thick 3Y-Z crowns cemented with RMGIC aside from 1.2 mm 5Y-Z crowns bonded with resin cement.

Only one previous study also compared the strength of 3Y-Z and 5Y-Z crowns at different thicknesses. The study from Adabo et al. of particle-abraded crowns bonded to metallic dies reported that 1.5 mm 5Y-Z premolar crowns had a lower characteristic strength than 1.0 mm 3Y-Z crowns [12]. In the current study, 1.2 mm bonded premolar 5Y-Z crowns had a similar strength to 1.0 mm 3Y-Z cemented crowns. The difference in the results is because a resin cement was used for all groups in their study, whereas the 3Y-Z control groups were cemented with RMGIC in the current study.

In a study by Alraheam et al., 30% of 1.2 mm 5Y-Z discs failed in fatigue, whereas 0% of the 0.7 mm 3Y-Z discs failed in fatigue [12]. Another study by Machry et al. of bonded, particle-abraded discs reported that 1.0 mm discs of 5Y-Z bonded to epoxy resin had a lower fracture resistance than 0.7 mm discs of 3Y-Z bonded to epoxy dies [15]. These studies suggest that a 1.0 mm thickness of 5Y-Z is not as strong as 0.7 mm of 3Y-Z. As a result, a thickness greater than 1.0 mm was suggested for 5Y-Z. These results vary slightly from the current study in which 0.8 mm thick 5Y-Z was equivalent to 0.5 mm 3Y-Z controls. The current study differs from these previous studies as crowns were used rather than discs, RMGIC was used for 3Y-Z controls, and the 3Y-Z was used at its manufacturer’s recommended minimum thickness of 0.5 mm.

A study from Longhini et al. reported a similar fracture resistance of 0.5 mm particle-abraded and resin-bonded discs of 5Y-Z and 3Y-Z [14]. Additionally, the 1.5 mm bonded 5Y-Z discs had a similar strength to the 1.0 mm 3Y-Z bonded discs. The conclusions of their study vary from the current study due to the use of resin cement for the 3Y-Z control group.

The manufacturer’s recommended thickness for 5Y-Z is 1 mm, and no cementation type is specified [11]. Although the present study suggests that 5Y-Z can be used at 0.8 mm with RMGIC, it is prudent to follow the manufacturer’s instructions of a 1 mm minimum thickness. In order to achieve a fracture resistance equivalent to a 1 mm 3Y-Z crown (and a 1.5 mm metal ceramic crown), 5Y-Z would need to be 1.2 mm and bonded with resin cement.

Air particle abrasion is recommended for the preparation of zirconia crowns in the instructions for use of both the resin and RMGI cements used in this study [29]. In a previous study, air particle abrasion did not significantly decrease the fracture resistance of 5Y-Z crowns cemented with RMGIC [18]. These results are in contrast with studies which report that air particle abrasion reduces the strength of 5Y-Z discs [22,25,26]. As air particle abrasion has not been shown to impact the bond strength with RMGIC, it is at the clinicians’ and laboratories’ discretion if 5Y-Z crowns are routinely air particle-abraded [27].

To substitute for tooth structure, this study used a 3D-printed resin die material similar to the frequently used epoxy resin in previous studies. The modulus and strength of printed resin dies (elastic modulus = 2.1 GPa, flexural strength = 97.1 MPa) are lower than dentin (elastic modulus = 19.64 GPa, flexural strength = 164.3 MPa), and their appropriateness as a die substitute could be questioned [38,40]. Pilot testing in our laboratory revealed that the fracture force of zirconia crowns on dentin dies was similar to the 3D-printed resin composite used in this study. The advantage of using 3D-printed dies rather than natural tooth dies is that the same geometry could be used for all specimens, whereas natural tooth specimens would vary in size and preparation design. A previous study revealed that zirconia crown fracture resistance testing was more discerning between groups with the use of resin dies rather than metal dies [15]. Another study comparing the fracture strength of zirconia crowns reported that crowns fractured on resin dies produced a similar strength to enamel dies, whereas metal dies produced a significantly higher strength [35].

The loading configuration used in the current study employs a rounded sphere loading a single cusp at 30° off the axis of the tooth (Figure 2). Likely load was transferred to the occlusal groove in some specimens as evidenced by the observation of the fracture surfaces of the crowns near the occlusal groove (Figure 4). A previous finite element analysis of crowns indented by a sphere reported that loading on steeper cusps transfers stress to the occlusal groove, whereas loading a flatter cusp concentrates stress below the indenter [38].

Several limitations could be observed in the present study. The results of this study are specific to the materials tested; differences in the composition of other brands of cement and ceramic materials may have significant effects on the results of the study. This study did not examine a group of 5Y-Z that was bonded with resin cement without air particle abrasion. That decision was made as the elimination of particle abrasion would significantly decrease the bond between 5Y-Z and resin cement and, therefore, eliminate the clinical advantage of the use of resin cement [22].

A blatant limitation of the current study is that crowns were tested in static load-to-failure testing rather than fatigue. High static loads applied by blunt spherical indenters produce Hertzian cone cracks located at the contact surface just outside the contact area. When lower stresses are applied cyclically, a single crack may be initiated on the cement surface which is driven towards the contact surface until fracture occurs. The latter form of fracture is more representative of clinical failures [33]. Fatigue studies may also incorporate a horizontal slide, which may transfer forces to the cement, leading to gaps [41]. Fatigue studies have limitations, including the technical challenge of discerning true failure when using a crown geometry and the large number of specimens and time required to generate an S-N curve. A future study might validate the relevant groups from the current study using fatigue cycling.

## 5. Conclusions

Based on the findings of this in vitro study and for the materials included in this study, the following conclusions were drawn:5Y-Z crowns as thin as 0.8 mm (regardless of the cement or surface treatment) have a similar fracture resistance to 0.5 mm thick 3Y-Z cemented with RMGIC. Despite this finding, manufacturers’ recommendations should be followed regarding the minimum restoration thickness.5Y-Z crowns with 1.2 mm thickness that are bonded with resin cement have a similar fracture resistance to 1.0 mm thick 3Y-Z cemented with RMGIC.

## Figures and Tables

**Figure 1 materials-17-00365-f001:**
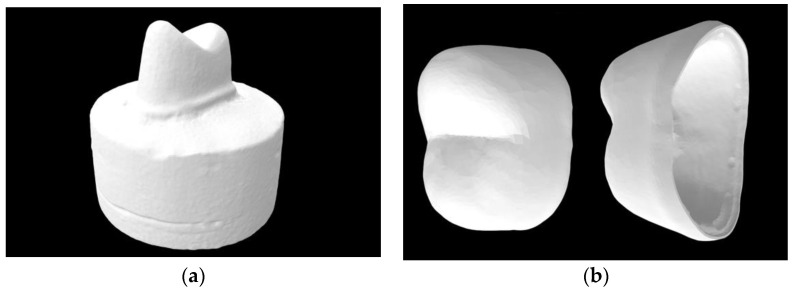
(**a**) Standardized tooth preparation; (**b**) representative zirconia crown design.

**Figure 2 materials-17-00365-f002:**
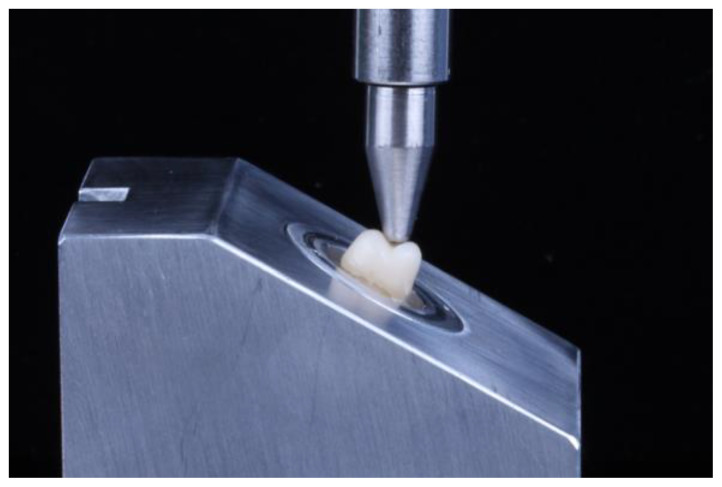
Loading of crowns at 30° off axis by 3.5 mm diameter steel indenter.

**Figure 3 materials-17-00365-f003:**
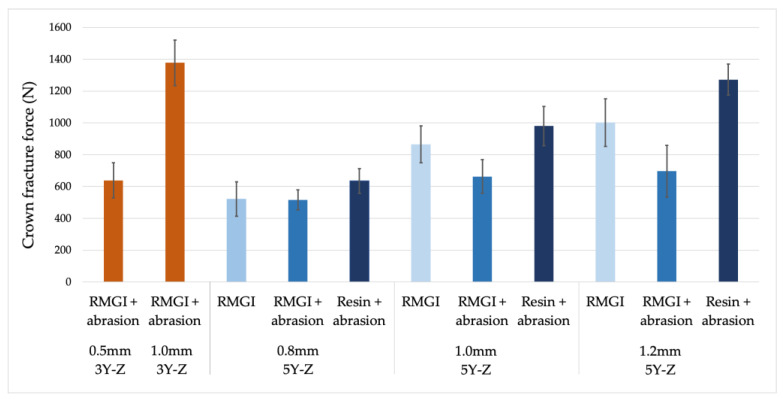
Crown fracture force (mean ± standard deviation).

**Figure 4 materials-17-00365-f004:**
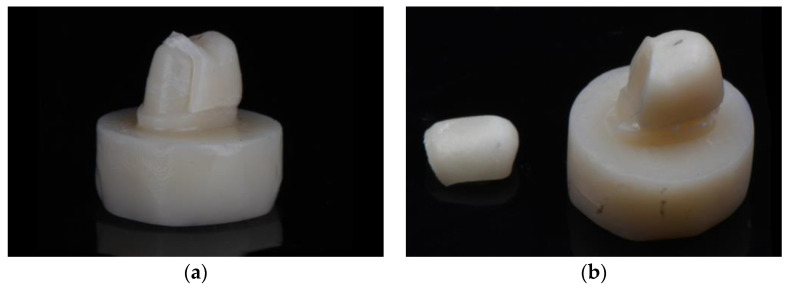
(**a**) Fracture of die alone; (**b**) fracture of die and crown.

**Table 1 materials-17-00365-t001:** Materials used in this study.

Material	Manufacturer	Composition
Katana UTML	Kuraray Noritake	5Y-Z
Katana HT	Kuraray Noritake	3Y-Z
Rely X Luting Cement	3M	RMGIC
Panavia SA Cement Universal	Kuraray Noritake	Resin cement

**Table 2 materials-17-00365-t002:** Crown fracture force (mean ± standard deviation).

Zirconia	Thickness (mm)	Particle Abrasion	Cement	Crown Fracture Force (N)
3Y-Z	0.5	Yes	RMGI	639.30 ± 111.77
	1.0	Yes	RMGI	1378.10 ± 143.22
5Y-Z	0.8	No	RMGI	522.67 ± 108.57 *
		Yes	RMGI	516.86 ± 63.32 *
		Yes	Resin	635.89 ± 78.00 *
	1.0	No	RMGI	865.30 ± 116.39 *
		Yes	RMGI	663.78 ± 106.80 *
		Yes	Resin	980.10 ± 123.50 *
	1.2	No	RMGI	1002.56 ± 149.58 *
		Yes	RMGI	696.30 ± 165.72 *
		Yes	Resin	1272.44 ± 97.78

* groups with a * were determined to be significantly lower than the 1.0 mm 3Y-Z control group. None of the groups were statistically lower than the 0.5 mm 3Y-Z control group.

## Data Availability

All source data may be obtained from the corresponding author.

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
