# Peer review of "The Effect of Restoration Thickness on the Fracture Resistance of 5 mol% Yttria-Containing Zirconia Crowns"

_materials, 2024, doi:10.3390/ma17020365_

Round 1

Reviewer 1 Report

Comments and Suggestions for Authors

In this work, the authors investigated the effect of restoration thickness on the flexural performance of 5 mol% yttria zirconia crown. In general, this is a well conducted and easy-to-read study, which adds to the body of knowledge available in this area of research, although major revisions as per the following report should be performed for both technical and language clarity, before the next submission. The following major comments might help to improve the quality of the manuscript:

1. The current version of the Introduction part is flawed. In the introduction part, the author is required to explain and emphasize the scientific novelty of this research through a comprehensive literature review of the subject. Besides, what’s the contribution of the present study to the current research area? The authors are encouraged to supplement some recent related references to support this description. There is recent publication recommended to add in the reference section: H.Q. Qiu, Y.Q. Zhang, W.W Huang, et al. Sintering properties of tetragonal zirconia nanopowder preparation of the NaCl+KCl binary system by the sol-gel-flux method. ACS Sustainable Chemistry & Engineering, 2023, 11(3): 1067-1077.

2. “Al2O3”? Please correct the chemical formula subscript. Please check for similar mistakes and revised carefully.

3. “37°C”? There should be a space between the number and the temperature unit (℃). Please check the entire text for similar errors and correct them carefully.

4. Why is the thickness not consistent when comparing 3Y-Z and 5Y-Z in Table 1? This seems to lack comparability, making it difficult for readers to understand. It is recommended that the thicknesses of the two be standardized for comparison.

5. It is recommended that the results be presented in the same chapter as the discussion instead of separating them.

6. Please do not use numbering in the conclusions part; instead, use point or dash line.

7. Check the format of references in consistent. Please carefully read the submission guidelines of the Journal, and revise the format and keep in consistent. For example, distinguish between uppercase and lowercase letters and the form of giving the number of authors.

I hope these suggestions could be well accepted by the authors.

Comments on the Quality of English Language

Moderate editing of English language required

Author Response

Thank you for the reviewer’s helpful comments.

  1. The current version of the Introduction part is flawed. In the introduction part, the author is required to explain and emphasize the scientific novelty of this research through a comprehensive literature review of the subject. Besides, what’s the contribution of the present study to the current research area? The authors are encouraged to supplement some recent related references to support this description. There is recent publication recommended to add in the reference section: H.Q. Qiu, Y.Q. Zhang, W.W Huang, et al. Sintering properties of tetragonal zirconia nanopowder preparation of the NaCl+KCl binary system by the sol-gel-flux method. ACS Sustainable Chemistry & Engineering, 2023, 11(3): 1067-1077.

Thank you.  We have included the purpose of this study into the introduction to better explain the need for this work in the context of the published literature. We have added the reviewer’s suggested reference.  

  1. “Al2O3”? Please correct the chemical formula subscript. Please check for similar mistakes and revised carefully.

We have corrected this error and searched the paper for similar punctuation errors.

  1. “37°C”? There should be a space between the number and the temperature unit (℃). Please check the entire text for similar errors and correct them carefully.

We have corrected this error and searched the paper for similar punctuation errors.

  1. Why is the thickness not consistent when comparing 3Y-Z and 5Y-Z in Table 1? This seems to lack comparability, making it difficult for readers to understand. It is recommended that the thicknesses of the two be standardized for comparison.

Thank you for addressing this concern. The stated objective of the study was to determine what thickness of the newer material (5Y-Z) would provide an equivalent strength as a material more established in the dental field (3Y-Z) at the thickness recommended by its manufacturer and the thickness recommended in the literature.  Testing 5Y-Z at 0.5mm would be outside of the manufacturer’s recommendations for the material as well.   We have included a purpose statement in the introduction to explain the rationale for these groups.

  1. It is recommended that the results be presented in the same chapter as the discussion instead of separating them.

Thank you for the recommendation. We followed the guideline of the journal and the included template of the journal which included a Results and Discussion section.  

  1. Please do not use numbering in the conclusions part; instead, use point or dash line.

Thank you, we have replaced the numbers with dots.

  1. Check the format of references in consistent. Please carefully read the submission guidelines of the Journal, and revise the format and keep in consistent. For example, distinguish between uppercase and lowercase letters and the form of giving the number of authors.

Thank you.  We have corrected all titles to be “sentence case” and listed all authors (for citations with less than 10 authors) according to the journal guidelines.

Reviewer 2 Report

Comments and Suggestions for Authors

The paper “The effect of restoration thickness on fracture resistance of 5 mol% yttria-containing zirconia crowns “ aimed to investigate the fracture resistance of 5Y-Z crowns at 0.8mm, 1.0 mm, and 1.2mm thickness that were bonded with resin cement (with air-particle abrasion) or cemented with RMGIC (with and without air-particle abrasion) ussing a crown fracture test.

The covered topic is of high interest for the readers. Nevertheless some improvements are required before possible publication.

Please add a table with the investigated materials, their composition, manufacturer, lots, etc.

In the Introduction and in the discussion relevance should be given to possible influence of margin preparation design on the vertical marginal gap. The authors could support the sentence/paragraph with the following paper:

Comba A, Baldi A, Carossa M, et al. Post-Fatigue Fracture Resistance of Lithium Disilicate and Polymer-Infiltrated Ceramic Network Indirect Restorations over Endodontically-Treated Molars with Different Preparation Designs: An In-Vitro Study. Polymers (Basel). 2022;14(23):5084. Published 2022 Nov 23. doi:10.3390/polym14235084

Why the authors decided not to use natural teeth? Epoxy resin is standardized but does not have the same mechanical properties of natural teeth. The authors could refer also to the precision of different materials to the experimental abutment thus influencing mechanical testing. The authors could refer to: https://doi.org/10.3390/ma16062413

Were the data checked for normality with a specific test?

What are the possible future studies that can be conducted based on the results of the current study?

What are the limitations?

Author Response

Thank you for the reviewer’s helpful comments.

Please add a table with the investigated materials, their composition, manufacturer, lots, etc.

A table has been added with the materials, manufacturer and composition.  Unfortunately, we no longer have access to the LOT numbers of the materials used.

In the Introduction and in the discussion relevance should be given to possible influence of margin preparation design on the vertical marginal gap. The authors could support the sentence/paragraph with the following paper:

Comba A, Baldi A, Carossa M, et al. Post-Fatigue Fracture Resistance of Lithium Disilicate and Polymer-Infiltrated Ceramic Network Indirect Restorations over Endodontically-Treated Molars with Different Preparation Designs: An In-Vitro Study. Polymers (Basel). 2022;14(23):5084. Published 2022 Nov 23. doi:10.3390/polym14235084

We have included the reviewers suggested reference.

Why the authors decided not to use natural teeth? Epoxy resin is standardized but does not have the same mechanical properties of natural teeth. The authors could refer also to the precision of different materials to the experimental abutment thus influencing mechanical testing. The authors could refer to: https://doi.org/10.3390/ma16062413

Thank you for recognizing this concern.  I have highlighted the paragraph in the discussion which explains the limitations of natural teeth (variability) and mentions pilot testing in our lab to confirm that fracture strength of zirconia is similar on resin dies as dentin (this data is in preparation for a manuscript).  I have also included references which have compared using different die materials for the fracture strength of zirconia crowns. 

Were the data checked for normality with a specific test?

Thank you for checking this.  We use a histogram to check and then verify with Shapiro-Wilk.  This has been added.

What are the possible future studies that can be conducted based on the results of the current study?

Thank you for the comment. A future study may look at fatigue testing of these materials.  This information has been highlighted at end of Discussion section.

What are the limitations?

The limitations of the study included the limited brands of materials, the exclusion of some combination of surface treatments/cements, as well was the exclusion of fatiguing.  This information has been highlighted in the Discussion section.

Round 2

Reviewer 1 Report

Comments and Suggestions for Authors

the author have been revised and answered all the question.

Reviewer 2 Report

Comments and Suggestions for Authors

All required improvements have been provided.